# High Accurate Mathematical Tools to Estimate the Gravity Direction Using Two Non-Orthogonal Inclinometers

**DOI:** 10.3390/s21175727

**Published:** 2021-08-25

**Authors:** Daniele Mortari, Anthony Gardner

**Affiliations:** Aerospace Engineering, Texas A&M University, 746C H.R. Bright Building, 3141 TAMU, College Station, TX 77843-3141, USA; anthonygardner@tamu.edu

**Keywords:** inclinometer data processing, covariance analysis, geoid correction

## Abstract

This study provides two mathematical tools to best estimate the gravity direction when using a pair of non-orthogonal inclinometers whose measurements are affected by zero-mean Gaussian errors. These tools consist of: (1) the analytical derivation of the gravity direction expectation and its covariance matrix, and (2) a continuous description of the geoid model correction as a linear combination of a set of orthogonal surfaces. The accuracy of the statistical quantities is validated by extensive Monte Carlo tests and the application in an Extended Kalman Filter (EKF) has been included. The continuous geoid description is needed as the geoid represents the true gravity direction. These tools can be implemented in any problem requiring high-precision estimates of the local gravity direction.

## 1. Introduction

The direct measurement of gravity using inclinometers has a variety of practical applications. Tilt sensors, such as accelerometers or inclinometers, are the hardware of choice for measuring a planet’s gravitational field. While the gravity is perturbed on a global scale by other massive bodies (e.g., Moon, Sun), local perturbations to the gravity vector’s pointing direction depend on nearby mass distributions and anomalies, or a lack thereof. A detailed review of various kinds of solid tilt sensors with comparisons between the most typical tilt sensing techniques can be found in [1].

From a sensory perspective, inclinometers are commonly used to measure inclination angle magnitudes and structural deformations. Measured values are typically provided as a percentage or angular deflection with respect to a level reference surface, whose nominal orientation is perpendicular to the local gravity vector. Other classic inclinometer use cases include measuring the slope gradient during activities such as tunneling, de-watering and excavation, as well as those that require monitoring the integrity of the ground around a structure.

In land surveying and mapping, an inclinometer can provide a rapid measurement of the slope of a geographic feature [2] or be used for analyzing the magnitude, rate, direction, depth, and type of landslides [3]. Continuous monitoring of landslides is important for early warning purposes. In general, inclinometers can be used for continuous monitoring of any movement, providing valuable information for landslide autonomous geometry reconstruction [4]. In the oil and gas industries, inclinometers are used to measure the strike and dip of geologic formations [5] in order to detect oil and mineral deposits. In forestry, tree height measurements can be made by an inclinometer using standardized methods [6]. Major artillery guns may have an associated inclinometer used to facilitate the aiming of shells over long distances [7,8]. Permanently installed inclinometers are employed at major earthworks such as dams [9,10] to monitor the long-term stability of the structure. Inclinometers can also be used in robotics [11] and to measure bridge deflections [12] for bridge safety evaluation. Specifically, the authors of [13] developed and implemented a measuring system to assess railway bridges using measurements of the structural response to passing trains. Furthermore, low-cost tilt sensors are now commonplace in commercially available smartphones in order to make relative orientation estimates of the device.

Inclinometers are also used in more complex systems, such as in problems of estimating the position and orientation of a calibrated camera observing a set of *n* 3D points, known as Perspective-*n*-Point (PnP) problems [14], and in “Stellar Positioning Systems” [15,16,17], where the gravity direction along with celestial references can be used to estimate the geographical position in scenarios where real-time GPS information is unavailable (e.g., on Mars). Such systems can also be used on Earth as a backup system during instances of GPS jamming or spoofing.

Today, new types of inclinometers are emerging, such as the two-dimensional highly sensitive fiber-optic inclinometer proposed by [18], which can provide long-term continuous monitoring of inclinations or real-time feedback control of tilt angles, especially in harsh environments with violent temperature variations. Microelectromechanical (MEMS) inclination sensors are also being developed to enhance the properties of design, fabrication, and signal measurement precision [19].

Prior to being deployed in the field, inclinometers are efficiently calibrated in a laboratory setting. Because of vibrations and subsequent thermal fluctuations, the orientation of the inclinometer assembly degrades over time, requiring re-calibration methods to properly account for new measurement errors. Reference [20] proposes a new inclinometer assembly error calibration and horizontal image correction method utilizing plumb lines, where the horizontal correction is derived by a homography matrix.

The monitoring of possible road subsidence is proposed using a combination of theodolite measurements and inclinometers [21] to detect ground motion near critical infrastructure during excavation in the vicinity. The proposed method, validated by Monte Carlo simulation, is formulated as a covariance weighted Hermite approximation problem. On the contrary, this paper addresses the problem of the best estimation of the gravity direction by non-approximated (analytical) derivations for a pair of inclinometers only. The proposed method can then be adopted in almost all the scenarios involving two inclinometers.

This study considers the classic system of two non-orthogonal inclinometers affected by zero-mean Gaussian error (hypothesis) to measure the gravity direction, and it provides the non-approximated mathematical tools to best estimate the gravity direction for any static and/or dynamic scenario (objective). In particular, we provide:The analytical tools to best estimate the gravity direction. These tools consist of: (a) an exact estimate of the gravity direction when inclinometer measurements are affected by zero-mean Gaussian noise, and (b) the associated gravity covariance matrix. These tools can be used in static and filtered dynamic scenarios to accurately estimate the actual gravity direction and to model the local subtle gravity variations.A continuous mathematical model to describe the true gravity direction deviation with respect to the latest (1984) revision of the approximated World Geodetic System (WGS-84) ellipsoidal model. Two deflection correction models are proposed for the north–south and east–west deflection variations. These models are expressed using combinations of a set of *N* linearly independent orthogonal surfaces that are derived from Chebyshev orthogonal polynomials of the first kind.

These tools and models find relevance in a variety of gravity measurement tasks and associated use, especially in filtered systems, such as the one developed for geo-technical monitoring in [22].

### System Definition

The system under consideration in this study is shown in Figure 1, which describes the axes of the two non-orthogonal inclinometers, $\left[I_{x},I_{y}\right]$, and an orthonormal inclinometer reference frame, {x,y,z}. The $I_{x}$-axis of the inclinometer frame is assumed to be coincident with the x-axis reference frame, and the plane defined by the two inclinometers axes is also coincident with the [x,y]-plane. The relative orientation of the two inclinometer axes deviates from the nominal/optimal orthogonal displacement by an angle ε. This angle, which takes into account the small deviation from orthogonality due to mounting errors, can be estimated during lab calibration and, possibly, re-calibrated later to account for variations caused by vibrations and/or thermal expansions.

The gravity direction in the [x,y,z] frame is
(1)$$g=\left\{\begin{array}{c} g_{x}\\ g_{y}\\ g_{z} \end{array}\right\}=\left\{\begin{array}{c} \cos\vartheta_{x}\\ \frac{\cos\vartheta_{y}-\cos\vartheta_{x}\;\sin\varepsilon }{\cos\varepsilon }\\ -\sqrt{1-g_{x}^{2}-g_{y}^{2}} \end{array}\right\},$$
where the third component enforces the unit-norm constraint. The inclinometer measurement angles, $\vartheta_{x}$ and $\vartheta_{y}$ (see Figure 1), allow the estimation of the gravity direction in the reference frame.

## 2. Gravity Direction Estimation

This section contains the covariance analysis of the gravity direction estimated by two identical inclinometers, as shown in Figure 1. The analysis is performed for the generic case of a non-orthogonal mounting, where the inclinometer axes differ from orthogonality by the angle ε.

Let the two measured angles, $\vartheta_{x}$ and $\vartheta_{y}$, be affected by Gaussian noise
$$\vartheta_{x}∼N\left(\mu_{x},\sigma^{2}\right)\;\mathrm{and}\;\vartheta_{y}∼N\left(\mu_{y},\sigma^{2}\right).$$
where $\mu_{x}$ and $\mu_{y}$ are the mean values and $\sigma^{2}$ is the measurement variance, identical for the two inclinometers. The mean values, $\mu_{x}$ and $\mu_{y}$, constitute the two main components of the EKF state vector. The quantities ε and $\sigma^{2}$ are initially estimated by lab calibration methods. However, vibrations and/or thermal expansions will likely change their values over time. In this case, these parameters should be re-calibrated and their values re-estimated by adding them as new variables to the EKF state vector (see the Discussion section).

The angles, $\vartheta_{x}$ and $\vartheta_{y}$, can be represented by a zero-mean Gaussian angle, δ, as
$$\left\{\begin{array}{c} \vartheta_{x}=\mu_{x}+\delta \\ \vartheta_{y}=\mu_{y}+\delta \end{array}\right.,\;\mathrm{where}\;\delta ∼N\left(0,\sigma^{2}\right).$$

To estimate the gravity direction, Equation (Equation 1) requires the evaluation of $E\{\cos\vartheta_{x}\}$ and $E\{\cos\vartheta_{y}\}$. Dropping the axis-specific subscript, we can write,
(2)E{cosϑ}=E{cos(μ+δ)}=cosμ·E{cosδ}−sinμ·E{sinδ}.

The term sinδ is an odd function. Therefore, its expectation is zero. Hence, Equation (Equation 2) simplifies to
E{cosϑ}=cosμ·E{cosδ}.

To compute E{cosδ}, let us expand it by Maclaurin series [23]:(3)$$E\left\{\cos\delta \right\}=E\left\{\sum_{k=0}^{\infty }\frac{{\left(-1\right)}^{k}}{(2k)!}\delta^{2k}\right\}=\sum_{k=0}^{\infty }\frac{{\left(-1\right)}^{k}}{(2k)!}E\left\{\delta^{2k}\right\}.$$

Reference [24] provides the following identity:(4)$$E\left\{\delta^{2k}\right\}=1\cdot 3\cdot 5\cdot \ldots \cdot \left(2k-1\right)\;\sigma^{2k}=\frac{(2k)!}{2^{k}\;k!}\;\sigma^{2k}.$$

Substituting Equation (Equation 4) into Equation (Equation 3), the exact expectation of the cosine of the zero-mean Gaussian angle, δ, is
$$E\left\{\cos\delta \right\}=\sum_{k=0}^{\infty }\frac{{\left(-1\right)}^{k}}{2^{k}\;k!}\;\sigma^{2k}=\sum_{k=0}^{\infty }\frac{{\left(-1\right)}^{k}}{k!}{\left(\frac{\sigma^{2}}{2}\right)}^{k}=e^{-\sigma^{2}/2}.$$

Therefore, we obtain
(5)$$\left\{\begin{array}{c} E\left\{\cos\vartheta_{x}\right\}=e^{-\sigma^{2}/2}\;\cos\mu_{x}\\ E\left\{\cos\vartheta_{y}\right\}=e^{-\sigma^{2}/2}\;\cos\mu_{y} \end{array}\right..$$

This result allows us to find the expected values of the first two components of the gravity direction: (6)$$\begin{array}{cc} {\hat{g}}_{x} & =E\left\{g_{x}\right\}=e^{-\sigma^{2}/2}\;\cos\mu_{x} \end{array}$$(7)$$\begin{array}{cc} {\hat{g}}_{y} & =E\left\{g_{y}\right\}=e^{-\sigma^{2}/2}\left(\frac{\cos\mu_{y}}{\cos\varepsilon }-\cos\mu_{x}\;\tan\varepsilon \right). \end{array}$$

Figure 2 provides a geometrical interpretation of why the expectation of the cosine is lower than the cosine of the mean when the measured angle is affected by zero-mean error. If the inclinometer is not biased, then the error of the inclinometer can be described by a small cone whose axis is coincident with the inclinometer axis. This implies that inclinometer measurements affected by errors will fall within this cone. If the true angle with respect to the gravity direction is ϑ, then all measurements providing the true angle belong to the surface of a cone whose axis is the gravity direction and aperture is the angle ϑ. This cone intersects the measurement error cone by splitting it into two spherical areas that, as shown in Figure 2, are identified as “A” and “B.” All measurement angles falling in the “A” area provide an angle greater than the true value, while all the measurement angles falling into the “B” area provide an angle lower than the true value. Since the “A” area is greater than the “B” area, a zero-mean inclinometer will provide an angle greater than the true value. This means that *a zero-mean inclinometer provides biased angles.* This phenomenon is generated by our three-dimensional spherical world, and only vanishes under two situations: (a) when the measurement angle is perfectly $\vartheta =90^{\circ }$, and (b) when the inclinometer standard deviation (dictating the aperture of the measurement error cone) is $\sigma =0^{\circ }$ (idealized inclination sensor).

This bias effect is actually very small, even for poorly accurate commercial inclinometers. However, since the inclinometer axis direction is normally placed perpendicular to the gravity direction, this effect is even smaller, because the inclinometer works around the nominal orthogonality displacement of $\vartheta =90^{\circ }$.

## 3. Gravity Direction Covariance Matrix

The gravity direction covariance matrix is
$$P=E\left\{\left(g-\hat{g}\right){\left(g-\hat{g}\right)}^{T}\right\}=\left[\begin{array}{ccc} E\left\{g_{x}^{2}\right\}-{\hat{g}}_{x}^{2} & E\left\{g_{x}g_{y}\right\}-{\hat{g}}_{x}\;{\hat{g}}_{y} & E\left\{g_{x}g_{z}\right\}-{\hat{g}}_{x}\;{\hat{g}}_{z}\\ \left(\mathrm{sym}\right) & E\left\{g_{y}^{2}\right\}-{\hat{g}}_{y}^{2} & E\left\{g_{y}g_{z}\right\}-{\hat{g}}_{y}\;{\hat{g}}_{z}\\ \left(\mathrm{sym}\right) & \left(\mathrm{sym}\right) & E\left\{g_{z}^{2}\right\}-{\hat{g}}_{z}^{2} \end{array}\right].$$

To compute the P(1,1), P(1,2), and P(2,2) terms of this matrix, the following estimations are required:(8)$$\left\{\begin{array}{c} E\left\{g_{x}^{2}\right\}=E\left\{\cos^{2}\vartheta_{x}\right\}\\ E\left\{g_{x}\;g_{y}\right\}=E\left\{\cos\vartheta_{x}\;\cos\vartheta_{y}\right\}\frac{1}{\cos\varepsilon }-E\left\{\cos^{2}\vartheta_{x}\right\}\tan\varepsilon \\ E\left\{g_{y}^{2}\right\}=\frac{E\left\{\cos^{2}\vartheta_{y}\right\}-2E\left\{\cos\vartheta_{x}\;\cos\vartheta_{y}\right\}\sin\varepsilon +E\left\{\cos^{2}\vartheta_{x}\right\}\sin^{2}\varepsilon }{\cos^{2}\varepsilon } \end{array}\right.$$

The term $E\{\cos\vartheta_{x}\;\cos\vartheta_{y}\}$ can be expressed as
(9)$$E\left\{\cos\vartheta_{x}\cos\vartheta_{y}\right\}=E\left\{\cos\vartheta_{x}\right\}\;E\left\{\cos\vartheta_{y}\right\}=\cos\mu_{x}\;\cos\mu_{y}\;e^{-\sigma^{2}},$$
because $\cos\vartheta_{x}$ and $\cos\vartheta_{y}$ are statistically independent, while the computation of the $E\{\cos^{2}\vartheta \}$ terms requires the estimation of $E\{\cos^{2}\delta \}$ and $E\{\sin^{2}\delta \}$. We have
$$E\left\{\cos^{2}\vartheta \right\}=E\left\{{\left(\cos\mu \;\cos\delta -\sin\mu \;\sin\delta \right)}^{2}\right\}=\cos^{2}\mu \;E\left\{\cos^{2}\delta \right\}+\sin^{2}\mu \;E\left\{\sin^{2}\delta \right\}.$$

The $E\{\cos^{2}\delta \}$ and $E\{\sin^{2}\delta \}$ expectations can also be computed by Maclaurin series:$$\begin{array}{ccc} E\{\cos^{2}\delta \}\;\; & = & \;\;E\left\{\frac{1}{2}+\frac{1}{2}\sum_{k=0}^{\infty }\frac{{\left(-1\right)}^{k}2^{2k}}{(2k)!}\;\delta^{2k}\right\}=\frac{1}{2}+\frac{1}{2}\sum_{k=0}^{\infty }\frac{{\left(-1\right)}^{k}2^{2k}}{(2k)!}\;E\left\{\delta^{2k}\right\}\\ & = & \frac{1}{2}+\frac{1}{2}\sum_{k=0}^{\infty }\frac{{\left(-1\right)}^{k}\;2^{2\;k}}{(2\;k)!}\cdot \frac{(2k)!}{2^{k}\;k!}\cdot \sigma^{2k}=\frac{1}{2}+\frac{1}{2}\sum_{k=0}^{\infty }\frac{{\left(-1\right)}^{k}\;2^{k}}{k!}\cdot \sigma^{2k}\\ & = & \frac{1}{2}\;\left(1+e^{-2\;\sigma^{2}}\right) \end{array}$$
and, similarly,
$$E\left\{\sin^{2}\delta \right\}=E\left\{\frac{1}{2}-\frac{1}{2}\sum_{k=0}^{\infty }\frac{{\left(-1\right)}^{k}2^{2k}}{(2k)!}\;\delta^{2k}\right\}=\cdots =\frac{1}{2}\left(1-e^{-2\;\sigma^{2}}\right).$$

Therefore,
$$E\left\{\cos^{2}\vartheta \right\}=\frac{\cos^{2}\mu }{2}\left(1+e^{-2\;\sigma^{2}}\right)+\frac{\sin^{2}\mu }{2}\left(1-e^{-2\;\sigma^{2}}\right)=\frac{2\cos^{2}\mu -1}{2}\;e^{-2\;\sigma^{2}}+\frac{1}{2},$$
and, specifically,
(10)$$\left\{\begin{array}{c} E\left\{\cos^{2}\vartheta_{x}\right\}=\frac{2\cos^{2}\mu_{x}-1}{2}\;e^{-2\;\sigma^{2}}+\frac{1}{2}\\ E\left\{\cos^{2}\vartheta_{y}\right\}=\frac{2\cos^{2}\mu_{y}-1}{2}\;e^{-2\;\sigma^{2}}+\frac{1}{2} \end{array}\right..$$

Equations (Equation 9) and (Equation 10) allow us to obtain the following expectation expressions: (11)$$\begin{array}{cc} E\{g_{x}^{2}\} & =\frac{2\cos^{2}\mu_{x}-1}{2}\;e^{-2\;\sigma^{2}}+\frac{1}{2} \end{array}$$(12)$$\begin{array}{cc} E\{g_{x}\;g_{y}\} & =\frac{\cos\mu_{x}\;\cos\mu_{y}\;e^{-\sigma^{2}}}{\cos\varepsilon }-\left[\frac{2\cos^{2}\mu_{x}-1}{2}\;e^{-2\;\sigma^{2}}+\frac{1}{2}\right]\tan\varepsilon \end{array}$$(13)$$\begin{array}{cc} E\{g_{y}^{2}\} & =\frac{1}{\cos^{2}\varepsilon }\left[\frac{2\cos^{2}\mu_{y}-1}{2}\;e^{-2\;\sigma^{2}}+\frac{1}{2}\right]-\cos\mu_{x}\;\cos\mu_{y}\;e^{-\sigma^{2}}\;\frac{2\sin\varepsilon }{\cos^{2}\varepsilon }\\ \; & \;+\left[\frac{2\cos^{2}\mu_{x}-1}{2}\;e^{-2\;\sigma^{2}}+\frac{1}{2}\right]\tan^{2}\varepsilon \end{array}$$

Using Equations (Equation 6) and (Equation 11), the term, $P\left(1,1\right)=E\left\{g_{x}^{2}\right\}-{\hat{g}}_{x}^{2}$, of the covariance matrix can be written as
$$P\left(1,1\right)=\frac{2\cos^{2}\mu_{x}-1}{2}\;e^{-2\;\sigma^{2}}+\frac{1}{2}-{\left(e^{-\sigma^{2}/2}\;\cos\mu_{x}\right)}^{2}$$

This expression can be simplified into the following more compact form:(14)$$P\left(1,1\right)=\left(\frac{1-e^{-2\;\sigma^{2}}}{2}\right)-\left(e^{-\sigma^{2}}-e^{-2\;\sigma^{2}}\right)\cos^{2}\mu_{x}$$

To derive the analytical expression of the term, $P\left(2,2\right)=E\left\{g_{y}^{2}\right\}-{\hat{g}}_{y}^{2}$, we make use of Equations (Equation 7) and (13):$$\begin{array}{cc} P(2,2) & =\frac{1}{\cos^{2}\varepsilon }\left[\frac{2\cos^{2}\mu_{y}-1}{2}\;e^{-2\;\sigma^{2}}+\frac{1}{2}\right]-\cos\mu_{x}\;\cos\mu_{y}\;e^{-\sigma^{2}}\;\frac{2\sin\varepsilon }{\cos^{2}\varepsilon }\\ \; & \;+\left[\frac{2\cos^{2}\mu_{x}-1}{2}\;e^{-2\;\sigma^{2}}+\frac{1}{2}\right]\tan^{2}\varepsilon -e^{-\sigma^{2}}{\left(\frac{\cos\mu_{y}}{\cos\varepsilon }-\cos\mu_{x}\;\tan\varepsilon \right)}^{2} \end{array}$$

After some simple manipulations and simplifications, the previous equation can be written in the more compact form
(15)$$P\left(2,2\right)=\frac{1+\sin^{2}\varepsilon }{\cos^{2}\varepsilon }\left(\frac{1-e^{-2\;\sigma^{2}}}{2}\right)+\frac{\cos^{2}\mu_{x}\sin^{2}\varepsilon -\cos^{2}\mu_{y}}{\cos^{2}\varepsilon }\left(e^{-\sigma^{2}}-e^{-2\;\sigma^{2}}\right)$$

Note that, as a sanity check, if the two inclinometers were perfectly orthogonal, meaning if ε=0, then Equation (Equation 15) becomes formally identical to Equation (Equation 14).

As for the term $P\left(1,2\right)=E\left\{g_{x}g_{y}\right\}-{\hat{g}}_{x}\;{\hat{g}}_{y}$, we have
$$\begin{array}{cc} P(1,2) & =\frac{\cos\mu_{x}\;\cos\mu_{y}\;e^{-\sigma^{2}}}{\cos\varepsilon }-\left[\frac{2\cos^{2}\mu_{x}-1}{2}\;e^{-2\;\sigma^{2}}+\frac{1}{2}\right]\tan\varepsilon \\ \; & \;-e^{-\sigma^{2}}\left(\frac{\cos\mu_{x}\;\cos\mu_{y}}{\cos\varepsilon }-\cos^{2}\mu_{x}\;\tan\varepsilon \right) \end{array}$$
which can be simplified to
(16)$$P\left(1,2\right)=\left[\left(e^{-\sigma^{2}}-e^{-2\;\sigma^{2}}\right)\cos^{2}\mu_{x}-\left(\frac{1-e^{-2\;\sigma^{2}}}{2}\right)\right]\tan\varepsilon ,$$

This equation highlights an unexpected result: *the covariance term, P(1,2), is not a function of $\mu_{y}$.* This result, which has been numerically verified by extensive Monte Carlo tests, remains with no explanation. Note also that, if the two inclinometers were perfectly orthogonal (ε=0), then we would obtain P(1,2)=0.

### Computation of the P(1,3), P(2,3) and P(3,3) Terms via Covariance Law

The analytical expressions for the three missing terms of the covariance matrix, P(1,3), P(2,3) and P(3,3), were not found. However, a highly accurate estimation of these terms can be obtained using the covariance law [25,26], sometimes called the error propagation law.

The covariance law connects the two covariance matrices that are expressed in terms of two different (but equivalent) parameterizations by the Jacobian, J, of the transformation between parameterizations. In this specific case, the covariance law connects the 3×3 covariance matrix of the gravity unit-vector, *P*, with the 2×2 covariance matrix written in terms of the cosine of the measurements, *C*. This relationship consists of the identity
(17)$$P\approx J\;C\;J^{T}$$
where the analytical expressions of the covariance matrix terms,
$$C=\left[\begin{array}{ccc} E\left\{\cos^{2}\vartheta_{x}\right\}-E{\left\{\cos\vartheta_{x}\right\}}^{2} & \; & E\left\{\cos\vartheta_{x}\cos\vartheta_{y}\right\}-E\left\{\cos\vartheta_{x}\right\}E\left\{\cos\vartheta_{y}\right\}\\ \left(\mathrm{sym}\right) & \; & E\left\{\cos^{2}\vartheta_{y}\right\}-E{\left\{\cos\vartheta_{y}\right\}}^{2} \end{array}\right]$$
can all be computed using Equations (Equation 5), (Equation 9) and (Equation 10), while the Jacobian of the transformation between the two parameterizations is
$$J=\left[\begin{array}{ccc} \frac{\partial g_{x}}{\partial \cos\vartheta_{x}} & \; & \frac{\partial g_{x}}{\partial \cos\vartheta_{y}}\\ \frac{\partial g_{y}}{\partial \cos\vartheta_{x}} & \; & \frac{\partial g_{y}}{\partial \cos\vartheta_{y}}\\ \frac{\partial g_{z}}{\partial \cos\vartheta_{x}} & \; & \frac{\partial g_{z}}{\partial \cos\vartheta_{y}} \end{array}\right].$$

The first four elements of the Jacobian can be computed from Equation (Equation 1). Their expressions are
$$\left\{\begin{array}{c} \frac{\partial g_{x}}{\partial \cos\vartheta_{x}}=1,\\ \frac{\partial g_{x}}{\partial \cos\vartheta_{y}}=0 \end{array}\right.\;\mathrm{and}\;\left\{\begin{array}{c} \frac{\partial g_{y}}{\partial \cos\vartheta_{x}}=-\tan\varepsilon ,\\ \frac{\partial g_{y}}{\partial \cos\vartheta_{y}}=-\frac{1}{\cos\varepsilon } \end{array}\right.$$
while the last two elements can be computed using the chain rule as
$$\left\{\begin{array}{cc} \frac{\partial g_{z}}{\partial \cos\vartheta_{x}} & =\frac{\partial g_{z}}{\partial g_{x}}\cdot \frac{\partial g_{x}}{\partial \cos\vartheta_{x}}+\frac{\partial g_{z}}{\partial g_{y}}\cdot \frac{\partial g_{y}}{\partial \cos\vartheta_{x}}=\frac{\partial g_{z}}{\partial g_{x}}-\frac{\partial g_{z}}{\partial g_{y}}\;\tan\varepsilon \\ \frac{\partial g_{z}}{\partial \cos\vartheta_{y}} & =\frac{\partial g_{z}}{\partial g_{x}}\cdot \frac{\partial g_{x}}{\partial \cos\vartheta_{y}}+\frac{\partial g_{z}}{\partial g_{y}}\cdot \frac{\partial g_{y}}{\partial \cos\vartheta_{y}}=-\frac{\partial g_{z}}{\partial g_{y}}\cdot \frac{1}{\cos\varepsilon } \end{array}\right.$$
where
(18)$$\begin{array}{cc} \frac{\partial g_{z}}{\partial g_{x}} & =\frac{g_{x}}{\sqrt{1-g_{x}^{2}-g_{y}^{2}}}=\frac{\cos\vartheta_{x}\;\cos\varepsilon }{\sqrt{\cos^{2}\varepsilon -\cos^{2}\vartheta_{x}-\cos^{2}\vartheta_{y}+2\cos\vartheta_{x}\;\cos\vartheta_{y}\;\sin\varepsilon }} \end{array}$$
(19)$$\begin{array}{cc} \frac{\partial g_{z}}{\partial g_{y}} & =\frac{g_{y}}{\sqrt{1-g_{x}^{2}-g_{y}^{2}}}=\frac{\cos\vartheta_{y}-\cos\vartheta_{x}\;\sin\varepsilon }{\sqrt{\cos^{2}\varepsilon -\cos^{2}\vartheta_{x}-\cos^{2}\vartheta_{y}+2\cos\vartheta_{x}\;\cos\vartheta_{y}\;\sin\varepsilon }} \end{array}$$

Finally, the expression of the Jacobian of the transformation is
$$J=\left[\begin{array}{ccc} 1 & \; & 0\\ -\tan\varepsilon & \; & -\frac{1}{\cos\varepsilon }\\ \frac{\partial g_{z}}{\partial g_{x}}-\frac{\partial g_{z}}{\partial g_{y}}\;\tan\varepsilon & \; & -\frac{\partial g_{z}}{\partial g_{y}}\cdot \frac{1}{\cos\varepsilon } \end{array}\right]$$
where the expressions of the two partials, $\frac{\partial g_{z}}{\partial g_{x}}$ and $\frac{\partial g_{z}}{\partial g_{y}}$, are provided by Equation (Equation 18) and Equation (Equation 19), respectively.

The covariance law can actually be used to estimate all the elements of the covariance matrix. However, this law has been used here only to estimate the three missing elements of the covariance matrix, P(1,3), P(2,3) and P(3,3). The main reason is that the covariance law is based on a linear Taylor expansion of the transformation and is therefore an approximated method, even if it provides a highly accurate estimation of the covariance matrix. On the contrary, the terms P(1,1), P(2,2) and P(1,2), provided by the Equations (Equation 14)–(Equation 16), are analytically exact.

It is easy to derive them by setting Q=C(1,2)J(3,1)+C(2,2)J(3,2). In doing so, the expressions of the last three terms of the covariance matrix can be written as
(20)$$\left\{\begin{array}{c} P(1,3)=C(1,1)\;J(3,1)+C(1,2)\;J(3,2)\\ P(2,3)=J(2,1)\;P(1,3)+J(2,2)\;Q\\ P(3,3)=J(3,1)\;P(1,3)+J(3,2)\;Q \end{array}\right.$$

## 4. Covariance Analysis Numerical Validation

The numerical validation of the covariance matrix was performed by Monte Carlo tests using the R2016b version of MATLAB software. All of the tests were carried out using two identical inclinometers affected by zero-mean Gaussian error with standard deviation 1σ=0.1 deg., and with axes differing from orthogonality by the angle ε=5 deg. A set of $10^{7}$ Monte Carlo tests were performed on 1000 gravity directions, uniformly generated in a cone with 30 deg. (π/6 rad.) aperture around the nominal gravity direction $g={\left\{0,\;0,\;-1\right\}}^{T}$. The uniformly distributed gravity directions were generated as
$$g=\left\{\begin{array}{c} \sin\phi \;\cos\lambda \\ \sin\phi \;\sin\lambda \\ \cos\phi \end{array}\right\}\;\mathrm{where}\;\left\{\begin{array}{c} \lambda ∼U[0,2\pi ]\\ \cos\phi ∼U[-1,\cos(\pi -\pi /6)] \end{array}\right.$$
where U[a,b] indicates a uniform distribution in the [a,b] range. The gravity direction components, $\left(g_{x},g_{y},g_{z}\right)$, and the two angles, $\left(\vartheta_{x},\vartheta_{y}\right)$, generated by this distribution have the histograms reported in Figure 3.

Since the three terms P(1,1), P(1,2) and P(2,2) are mathematically correct (meaning, not approximated), the Monte Carlo numerical validation tests were performed simply to quantify the accuracy of the three elements of the covariance matrix, P(1,3), P(2,3) and P(3,3), whose expressions were obtained by the covariance law.

The validation was performed by comparing the three elements of the covariance matrix, T(1,3), T(2,3) and T(3,3), numerically estimated by the Monte Carlo tests, and their approximated analytical expressions, P(1,3), P(2,3) and P(3,3), obtained using Equation (Equation 20).

The top three plots of Figure 4 show the histograms of T(1,3), T(2,3) and T(3,3), while the bottom three plots show the histograms of the differences with the three terms estimated using Equation (Equation 20). The results of these Monte Carlo tests highlight the accurate estimation of the approximated terms, P(1,3), P(2,3) and P(3,3), computed via covariance law. The values of the terms T(1,3) and T(2,3), numerically estimated by the Monte Carlo tests, fall within the $\left[-10^{-6},+10^{-6}\right]$ range. The differences with the terms P(1,3), P(2,3), estimated via the covariance law, are in the $\left[-10^{-9},+10^{-9}\right]$ range, a difference better than three orders of magnitude with respect to the numerically estimated values. The same accuracy error level is experienced for the P(3,3) term with respect to the numerically estimated T(3,3) term, whose values fall within the $\left[0,+10^{-6}\right]$ range.

### 4.1. Extended Kalman Filter (EKF) Validation

The variable to estimate is represented by the gravity direction, i.e., by a three-component vector. However, since the third component of the gravity vector, $g_{z}$, is not independent (actually, it is derived from $g_{x}$ and $g_{y}$), then a reduced two-state EKF estimator must be developed and used. In order to quantify the accuracy gain obtained using the closed-form expression of the covariance matrix, a static scenario was selected to validate the EKF estimator. This choice does not depend on the dynamics under consideration or the accuracy of the adopted dynamical model. The EKF summary is provided as follows:State model:
$$x_{k}={\left\{\begin{array}{c} g_{x}\\ g_{y} \end{array}\right\}}_{k}=x_{k-1}+w_{k-1}$$Observation model:
$$y_{k}={\left\{\begin{array}{c} {\tilde{\vartheta }}_{x}\\ {\tilde{\vartheta }}_{y} \end{array}\right\}}_{k}=h\left(x_{k}\right)+v_{k}=\cos^{-1}\left[e^{\;\sigma^{2}/2}{\left\{\begin{array}{c} g_{k}\\ g_{k}\;\cos\varepsilon +g_{k}\;\sin\varepsilon \end{array}\right\}}_{k}\right]+v_{k}$$Initialization: Given initial state vector, $x_{0}^{-}$, and initial covariance matrix, $P_{0}^{-}$, and assuming no process noise, $Q_{k}=E\left\{w_{k}w_{k}^{T}\right\}=0_{2\times 2}$, the measurement’s covariance is
(21)$$R_{k}=E\left\{v_{k}v_{k}^{T}\right\}=\sigma^{2}\;I_{2\times 2}$$

The initial state, which is given by Equation (Equation 1), can be computed using the first measurements, ${\tilde{\vartheta }}_{x}$ and ${\tilde{\vartheta }}_{y}$,
(22)$$x_{0}^{-}=\left\{\begin{array}{c} \cos{\tilde{\vartheta }}_{x}\\ \frac{\cos{\tilde{\vartheta }}_{y}-\cos{\tilde{\vartheta }}_{x}\;\sin\varepsilon }{\cos\varepsilon } \end{array}\right\}=\left\{\begin{array}{c} x_{0}\\ y_{0} \end{array}\right\}$$
while the state covariance matrix is
(23)$$P_{k}^{-}=\left[\begin{array}{cc} P_{k}\left(1,1\right) & P_{k}\left(1,2\right)\\ P_{k}\left(1,2\right) & P_{k}\left(2,2\right) \end{array}\right]$$
where $P_{k}\left(1,1\right)$, $P_{k}\left(1,2\right)$ and $P_{k}\left(2,1\right)$, are provided by Equation (Equation 14), Equation (Equation 16) and Equation (Equation 15), respectively. Specifically, they can be written in terms of the state vector, $x_{k}^{-}$,
$$\begin{array}{ccc} P_{k}\left(1,1\right) & = & \left(e^{-2\;\sigma^{2}}-e^{-\sigma^{2}}\right)x_{k}^{2}+\frac{1}{2}\left(1-e^{-2\;\sigma^{2}}\right)\\ P_{k}\left(1,2\right) & = & \left[\left(e^{-\sigma^{2}}-e^{-2\sigma^{2}}\right)x_{k}^{2}-\frac{1}{2}\left(1-e^{-2\sigma^{2}}\right)\right]\tan\varepsilon \\ P_{k}\left(2,2\right) & = & \left(y_{k}^{2}+2\;x_{k}\;y_{k}\;\tan\varepsilon +2x_{k}^{2}\;\tan^{2}\varepsilon \right)\left(e^{-2\;\sigma^{2}}-e^{-\sigma^{2}}\right)+\\ \; & \; & +\left(\frac{2-\cos^{2}\varepsilon }{\cos^{2}\varepsilon }\right)\frac{1}{2}\left(1-e^{-2\;\sigma^{2}}\right) \end{array}$$

Predictor: Since the problem under analysis is static,
(24)$$\begin{array}{ccc} x_{k}^{+} & = & x_{k-1}^{-} \end{array}$$
(25)$$\begin{array}{ccc} P_{k}^{+} & = & P_{k-1}^{-} \end{array}$$Corrector:
(26)$$\begin{array}{ccc} K_{k} & = & P_{k}^{+}\;J_{h}^{T}\left(x_{k}\right){\left(J_{h}\left(x_{k}\right)\;P_{k}^{+}\;J_{h}^{T}\left(x_{k}\right)+R_{k}\right)}^{-1} \end{array}$$
(27)$$\begin{array}{ccc} x_{k}^{-} & = & x_{k}^{+}+K_{k}\left(y_{k}-h\left(x_{k}^{+}\right)\right) \end{array}$$
(28)$$\begin{array}{ccc} P_{k}^{-} & = & \left[I-K_{k}\;J_{h}\left(x_{k}\right)\right]\;P_{k}^{+} \end{array}$$
where the Jacobian, $J_{h}\left(x_{k}\right)$, of the observation is given by
(29)$$J_{h}\left(x_{k}\right)=\frac{\partial h\left(x_{k}\right)}{\partial x_{k}}=\left[\begin{array}{cc} -\frac{1}{\sqrt{1-x_{k}^{2}\;e^{\;\sigma^{2}}}} & 0\\ -\frac{\sin\varepsilon }{d_{k}} & -\frac{\cos\varepsilon }{d_{k}} \end{array}\right]\;e^{\;\sigma^{2}/2},$$
where
(30)$$d_{k}=\sqrt{1-e^{\;\sigma^{2}}{\left(x_{k}\;\sin\varepsilon +y_{k}\;\cos\varepsilon \right)}^{2}}.$$

### 4.2. Numerical Tests

The true direction of the gravity in the inclinometer reference frame has been selected as
$$g_{\mathrm{true}}=-\left\{\begin{array}{c} \sin\alpha \;\cos\beta \\ \sin\alpha \;\sin\beta \\ \cos\alpha \end{array}\right\}\;\mathrm{where}\;\left\{\begin{array}{c} \theta =30\;\deg\\ \beta =135\;\deg \end{array}\right.$$

The inclinometer measurements are corrupted by a zero-mean Gaussian noise with a standard deviation of σ=0.1 deg., which is a common value for commercial inclinometers. In addition, the deviation from orthogonality was selected as ε=2 deg.

Figure 5 shows the results obtained using 200 samples. In particular, the top-left figure shows for the $g_{x}$ component: (1) the noisy measurements (black markers), (2) the true value (solid black line), (3) the filtered estimates (solid blue line) and (4) the ±3σ error bounds (dotted red lines). The top-right figure shows the same quantities for the $g_{x}$ component.

The bottom left and right plots show the absolute errors for the $g_{x}$ and $g_{y}$ components, respectively. These error plots highlight, thanks to the analytically exact expressions of the covariance matrix terms, a very fast convergence of the EKF estimator.

## 5. Continuous Gravity Description

The irregular Earth shape (and mass distribution) has presented significant challenges to surveyers and cartographers throughout history. The early map makers considered the Earth to be a perfect sphere prior to Isaac Newton positing an oblate spheroid shape model. The axial–symmetric ellipsoid description of Earth, which has been formalized as the World Geodetic System (WGS-84) [27], is reliably used as the standard shape model for most of today’s GPS applications. The WGS-84 is defined by estimates of the Earth’s equatorial ($R_{E}$) and polar ($R_{P}$) radii, and a flattening factor (*f*)
$$\left\{\begin{array}{c} R_{E}=6,378,137\;m\\ R_{P}=6,356,752.3\;m \end{array}\right.\;\mathrm{and}\;f=\frac{R_{E}-R_{P}}{R_{E}}.$$

Describing a point on the Earth’s surface can be done by Cartesian coordinates (x,y,z) or by spherical coordinates, using geocentric (θ) or geodetic (ϕ) latitude and longitude (λ). Figure 6 shows the two definitions of latitude for an ellipsoid-shaped model.

The relationship between these two latitude definitions is [28]:(31)$$\tan\theta ={\left(1-f\right)}^{2}\;\tan\phi .$$

The greatest angular difference between geocentric and geodetic latitudes is, for the WGS-84 model, approximately 700 arcseconds, a difference that would introduce over 21 km of position error on Earth. However, even if the WGS-84 ellipsoidal model describes the gravity direction better than the sphere model, the actual gravity direction is dictated by the geoid model [29].

The geoid was first described by Gauss as the shape that the ocean’s surface would take if the only acceleration forces acting on it were Earth’s gravitational field and the rotational dynamics of the planet. In reality, this is still a simplification of the surface always orthogonal to the gravity direcion. The geoid is particularly useful for describing the effect that mass anomalies (such as mountain ranges or caverns) have on the local pointing direction of gravity, which can deviate from the reference ellipsoid by tens, hundreds or even thousands of arcseconds. Figure 7 shows the main parameters defining the relationship between the geoid surface and the WGS-84 ellipsoid. *P* is a point on the Earth’s surface that is located at altitude *h* normal to the ellipsoid and altitude *H* normal to the geoid. This point on the geoid surface is located at altitude *N* with respect to the WGS-84 ellipsoid. The angle between the true gravity direction (normal to the geoid and provided by the inclinometers) and the fictitious gravity direction (normal to WGS-84 reference model) is the geoid correction.

It is also worth pointing out that, unlike the WGS-84 axial–symmetric ellipsoid model, the geoid cannot be described by a simple equation because it is a function of the Earth’s mass distribution. However, the geometry of the geoid surface can be derived with respect to the WGS-84 model using the derivatives of the Earth’s gravitational potential. The Earth’s gravitational potential can be described in terms of spherical harmonics:(32)$$V\left(r,\phi ,\lambda \right)=\frac{GM}{r}\left\{1+\sum_{n=0}^{\infty }{\left(\frac{R_{E}}{r}\right)}^{n}\sum_{m=0}^{n}P_{n}^{m}\left(\sin\phi \right)\left[C_{nm}\cos\left(m\lambda \right)+S_{nm}\sin\left(m\lambda \right)\right]\right\},$$
where $GM=3.986013\times 10^{14}$ m${}^{3}$/s${}^{2}$ is the Earth gravitational constant, (r,ϕ,λ) are the spherical coordinates at which the evaluation is made, $P_{n}^{m}\left(\sin\phi \right)$ are the associated Legendre polynomials of degree *n*, order *m* and argument sinϕ, and $C_{nm}$ and $S_{nm}$ are spherical harmonic coefficients.

Taking normalized partial derivatives of Equation (Equation 32) with respect to the geodetic latitude and longitude, we obtain a mathematical definition for the north–south (δN) and east–west (δE) deflections of the gravity direction with respect to the WGS-84 ellipsoidal model:(33)$$\delta N=-\frac{1}{\gamma r}\cdot \frac{\partial V}{\partial \phi }\;\mathrm{and}\;\delta E=-\frac{1}{\gamma r\cos\phi }\cdot \frac{\partial V}{\partial \lambda }$$
where γ is a scalar representing the theoretical normal gravity, whose value and clear explanation are provided in Reference [30].

### 5.1. Continuous Description of Geoid

The analytical complexities associated with the infinite series of Equations (Equation 32) and (Equation 33) make them unable to evaluate the gravity direction without introducing truncation error. From satellite measurement data, a geoid model can be constructed in order to generate δE and δN deflection reference databases for a specified grid of points on the Earth’s surface, defined by geodetic latitude $\left(\phi_{k}\right)$ and longitude $\left(\lambda_{k}\right)$. To use the geoid database, interpolation between discontinuous points is needed. The accuracy of this interpolation is obtained at the expense of carrying a larger database (fine grid geoid description). In this study, a simple continuous geoid model is derived in order to use geoid corrections in specific locations and/or in dynamic situations.

The continuous geoid model is provided by expressing the δE and δN deflections as linear combinations of a set of *N* orthogonal surfaces:(34)$$\delta E\left(\phi ,\lambda \right)=\sum_{k=1}^{N}\alpha_{k}\;S_{k}\left(x\left(\phi \right),y\left(\lambda \right)\right)\;\mathrm{and}\;\delta N\left(\phi ,\lambda \right)=\sum_{k=1}^{N}\beta_{k}\;S_{k}\left(x\left(\phi \right),y\left(\lambda \right)\right),$$
where the $\alpha_{k}$ and $\beta_{k}$ are the unknown coefficients and the $S_{k}\left(x\left(\phi \right),y\left(\lambda \right)\right)$ are a set of orthogonal surfaces satisfying
$$\int_{-1}^{+1}\int_{-1}^{+1}S_{i}\left(x,y\right)\;S_{i}\left(x,y\right)\;dx\;dy=0,\;\mathrm{if}\;i\neq j.$$

These orthogonal surfaces are built using orthogonal polynomials, such as Chebyshev polynomials, $T_{k}\left(x\right)$, which are orthogonal in the x∈[−1,+1] range. This means that, when dealing with a specific geographical range of interest, $\phi \in [\phi_{\min},\phi_{\max}]$ and $\lambda \in [\lambda_{\min},\lambda_{\max}]$, a mapping between the geographical coordinates [ϕ,λ] and the Chebyshev variables [x,y] must be defined. The most simple mapping is the linear
$$x=\frac{\phi -\phi_{\min}}{\phi_{\max}-\phi_{\min}}\;\mathrm{and}\;y=\frac{\lambda -\lambda_{\min}}{\lambda_{\max}-\lambda_{\min}}.$$

Using Chebyshev orthogonal polynomials, the orthogonal surfaces can be defined as
$$S_{k}\left(x,y\right)=T_{i}\left(x\left(\phi \right)\right)\;T_{j}\left(y\left(\lambda \right)\right)\;\mathrm{where}\;i,j=0,1,2,\cdots$$
where any two orthogonal surfaces cannot be associated with the same [i,j] pair of Chebyshev polynomials.

Chebyshev orthogonal polynomials of the first kind can conveniently be computed in a recursive way. Starting with $T_{0}\left(x\right)=1$ and $T_{1}\left(x\right)=x$, the subsequent terms are computed by the recursive relationship,
(35)$$T_{k}\left(x\right)=2xT_{k-1}\left(x\right)-T_{k-2}\left(x\right),\;k=2,3,\cdots .$$

The problem of fitting the geoid data grids, $\delta E(\phi_{i},\lambda_{j})$ and $\delta N(\phi_{i},\lambda_{j})$, consists of two over-determined linear systems:$$\delta E\left(\phi_{i},\lambda_{j}\right)=\sum_{k=1}^{N}\alpha_{k}\;S_{k}\left(x\left(\phi_{i}\right),y\left(\lambda_{j}\right)\right)\;\mathrm{and}\;\delta N\left(\phi_{i},\lambda_{j}\right)=\sum_{k=1}^{N}\beta_{k}\;S_{k}\left(x\left(\phi_{i}\right),y\left(\lambda_{j}\right)\right).$$
where the total number of the geoid database points is $N_{p}>N$, for the problem to admit a solution. The previous equation describes two linear systems that can be written in matrix form:$$S_{E}\;\alpha =\xi \;\mathrm{and}\;S_{N}\;\beta =\eta ,$$
where the two coefficient vectors, α and β, can be computed by least squares:$$\alpha ={\left(S_{E}^{T}\;S_{E}\right)}^{-1}\;S_{E}^{T}\;\xi \;\mathrm{and}\;\beta ={\left(S_{N}^{T}\;S_{N}\right)}^{-1}\;S_{N}^{T}\;\eta .$$

This continuous description of the geoid is applied in the next subsection to a geographical region where the geoid gravity direction deviations are particularly significant.

### 5.2. Example of Continuous Geoid Description: Himalaya Region

The results of a numerical example of this procedure are shown in Figure 8 providing north–south and east–west deviations, in arcsec, of the gravity direction with respect to the direction normal to the WGS-84 ellipsoid surface. The region of interest selected is in the coordinates range $\left[\phi_{\min},\phi_{\max}\right]=\left[28.083^{\circ },28.875^{\circ }\right]$ and $\left[\lambda_{\min},\lambda_{\max}\right]=\left[83.667^{\circ },84.458^{\circ }\right]$.

The associated geoid database consists of a 10×10 grid ($N_{p}=100$ points), spanning a region of 2000 km ×2000 km. The number of orthogonal surfaces used for the continuous geoid model was N=55. This is also the number of unknowns (size of α and β vectors). The data points of the geoid grid are shown with black markers while the continuous models are indicated by the surfaces. The $L_{2}$ norms of the least-squares residuals are shown in Figure 9. The condition number of the matrix to invert (in the least-squares process), which is provided in Figure 10 as a function of the number of orthogonal surfaces, does not exceed 200.

By selecting orthogonal polynomials (instead of monomials), the condition number of the least-squares matrix to invert is reduced and, consequently, the numerical process becomes more robust and provides a more reliable numerical estimation of the unknown coefficient vectors, α and β.

## 6. Discussion

This study introduced the mathematical tools necessary for extracting a highly accurate estimation of the local gravity (plumb-line) direction from a pair of non-orthogonal inclinometers. The statistical analyses performed herein can be used to quantify the performance of static tilt sensors, since the measured direction of gravity is known to be normal to the reference geoid surface. The next section provides an example use case of these tools for high-precision continuous gravity direction estimates under static operating conditions.

The static Stellar Positioning System [15,16,17] introduces a framework for estimating the geographical position of a vehicle on the surface of a planetary body using a pair of inclinometers along with an atomic clock and a night-sky camera to observe celestial directions (e.g., stars, visible planets). This system, which can be used in GPS-denied scenarios or in locations where the GPS signal is not available (i.e., on Moon or Mars), assumes that a reliably accurate estimate of the local gravity direction is available, which in turn would require a geoid correction. The resulting mathematical estimation problem is nonlinear and, therefore, can be solved by iterative nonlinear least squares. The mathematical procedure requires computing the Jacobian of the gravity direction and, consequently, the first derivatives of Equation (Equation 35). The first derivatives can be obtained using the recursive equation,
$$\frac{\;dT_{k}\left(x\right)}{\;dx}=2\left(T_{k-1}\left(x\right)+x\frac{\;dT_{k-1}\left(x\right)}{\;dx}\right)-\frac{\;dT_{k-2}\left(x\right)}{\;dx}.$$

More accurate models will require the computation of the Hessian or even higher-order terms. The subsequent derivatives needed in these cases can be computed recursively:$$\frac{\;d^{n}T_{k}}{\;dx^{n}}=2\left(n\;\frac{\;d^{n-1}T_{k-1}}{\;dx^{n-1}}+x\;\frac{\;d^{n}T_{k-1}}{\;dx^{n}}\right)-\frac{\;d^{n}T_{k-2}}{\;dx^{n}},\;n=2,3,\cdots .$$

### Future Research Directions

The authors are planning to validate the proposed gravity direction estimation method using two non-orthogonal inclinometers by:Including the mounting angle deviation from orthogonality, ε, as an element of the EKF state vector. This would keep the system calibrated from variations of ε caused by vibrations and/or thermal expansions.Integrating the proposed system with an Inertial Measurement Unit (IMU) to extend the use of the static “Stellar Positioning System” to dynamical scenarios. Similar to the previous point, this will extend the EKF state vector to account for the variations in the transformation matrix between the inclinometer and IMU reference frames.Including in the EKF state estimator, for the static use “Stellar Positioning System,” the transformation matrix between the inclinometer and the night camera reference frames.

As for the continuous geoid model, the authors are planning to complete the analysis on the geoid north–south and east–west deflection estimates by:Investigating the convergence radius as a function of the geographical coordinates.Investigating the accuracy of the convergence as a function of the geoid database grid resolution.Including, in the current nonlinear position estimation algorithm, a divergence identification algorithm that re-initializes the estimation process using a new random guess selected within an admissible distance bound.

## 7. Conclusions

This study introduces a new approach to accurately estimate the gravity direction using a pair of non-orthogonal inclinometers affected by zero-mean Gaussian errors. The proposed approach estimates the gravity direction and its 3×3 covariance matrix, with the main three terms derived analytically. The accuracy of these statistical parameters is numerically validated by Monte Carlo simulation. The proposed method is then validated using an Extended Kalman Filter, which highlights a fast convergence to a very accurate solution.

Since the gravity direction is normal to the geoid surface, a continuous geoid model for the north–south and east–west corrections is provided as a linear combination of a set of orthogonal surfaces that are derived using Chebyshev polynomials of the first kind. An example application is provided for a specific region (centered in the Himalayas) to show the continuous description of the gravity direction deviations with respect to the WGS-84 ellipsoid model.

This covariance analysis, along with the continuous description of the geoid, can be applied to any static or dynamic scenarios involving the measurement of gravity using a two-axis inclinometer.

## Figures and Tables

**Figure 1 sensors-21-05727-f001:**
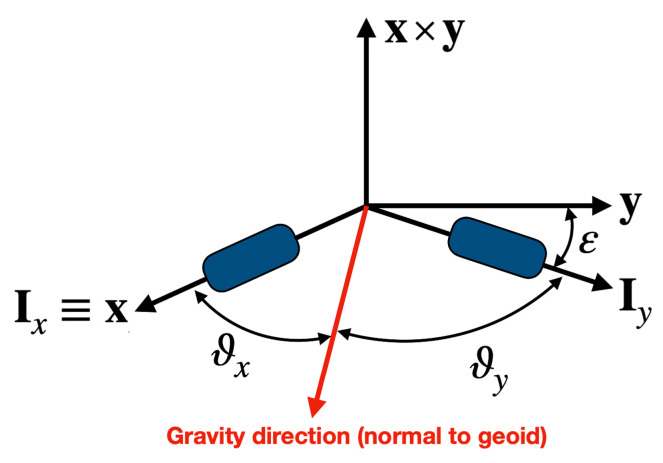
Geometry of the inclinometer system and associated reference frame.

**Figure 2 sensors-21-05727-f002:**
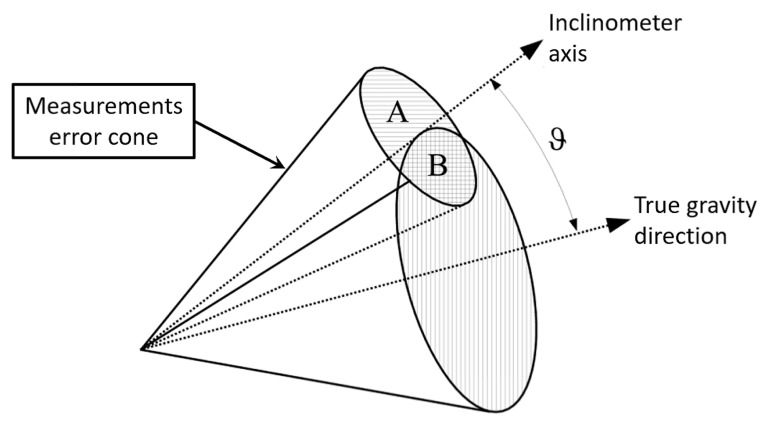
Geometrical explanation justifying the result provided by Equation (Equation 6).

**Figure 3 sensors-21-05727-f003:**
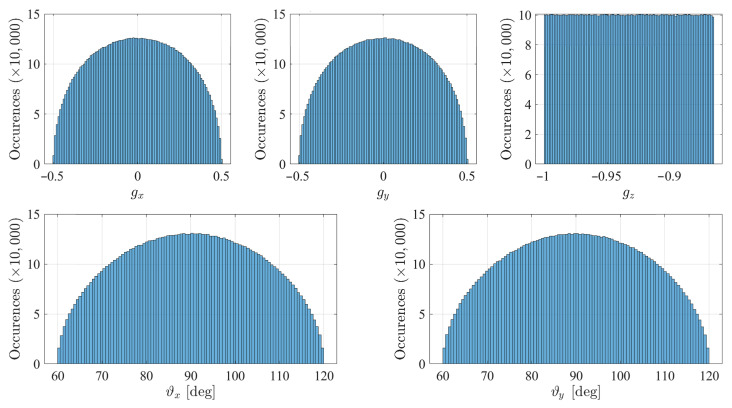
Monte Carlo tests: histograms of the angles and the gravity components.

**Figure 4 sensors-21-05727-f004:**
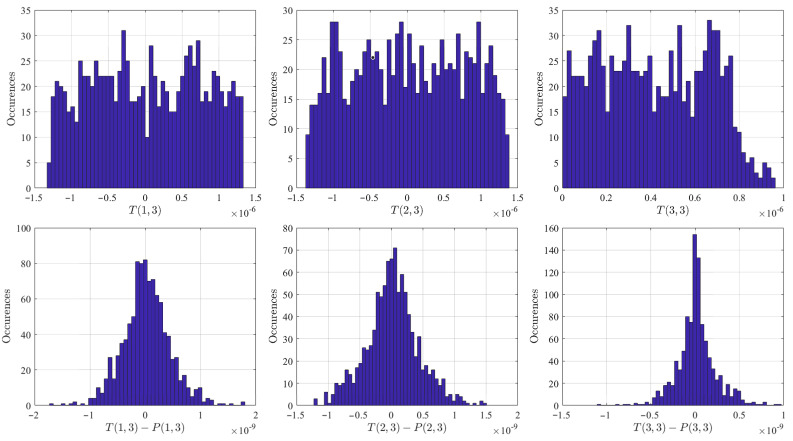
Monte Carlo validation for the P(1,3), P(2,3) and P(3,3) terms.

**Figure 5 sensors-21-05727-f005:**
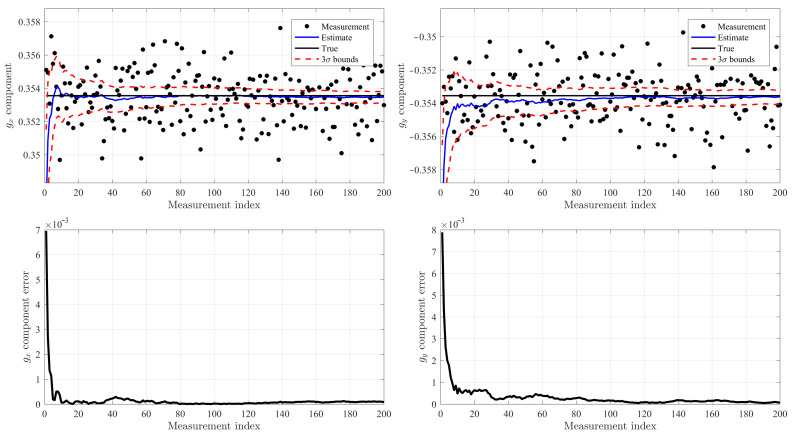
Extended Kalman Filter test results.

**Figure 6 sensors-21-05727-f006:**
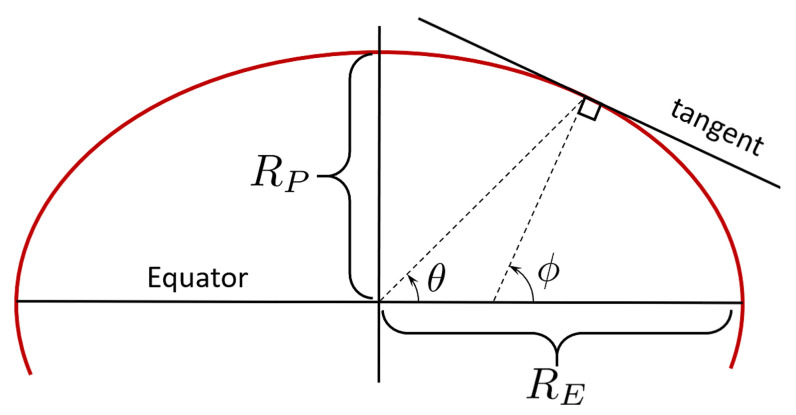
Geocentric (θ) and geodetic (ϕ) latitude angles.

**Figure 7 sensors-21-05727-f007:**
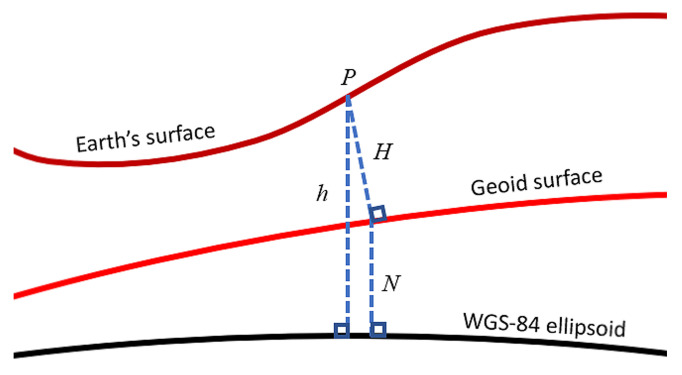
Gravity description with respect to geoid and reference ellipsoid surfaces.

**Figure 8 sensors-21-05727-f008:**
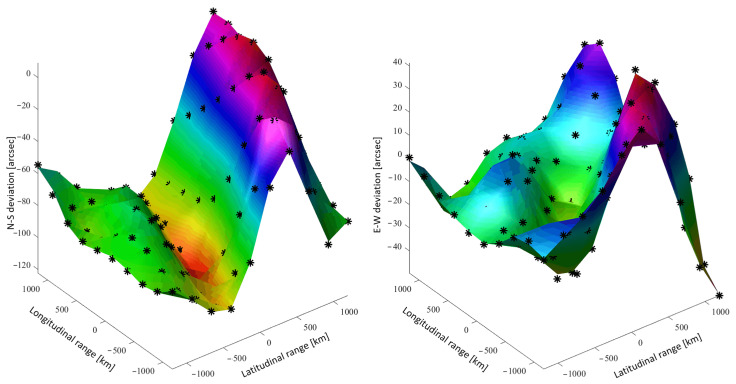
East–West and North–South geoid deviation (Himalaya region).

**Figure 9 sensors-21-05727-f009:**
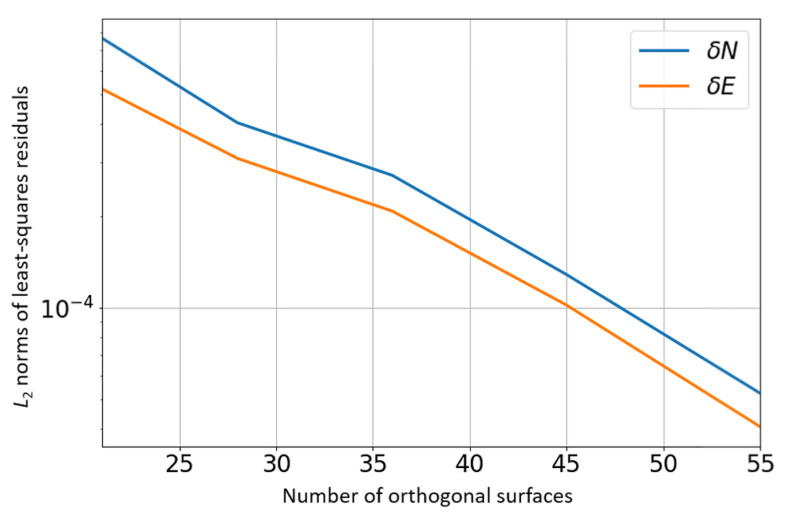
Residual $L_{2}$ norms.

**Figure 10 sensors-21-05727-f010:**
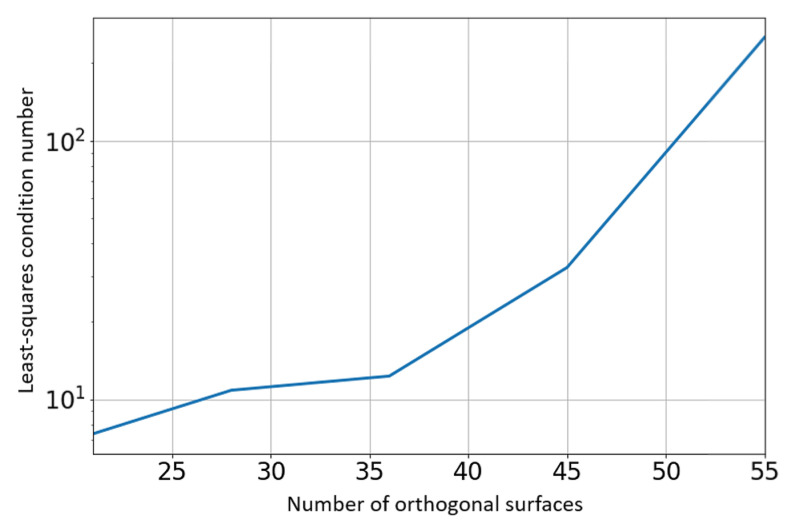
Least-squares condition number.

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
