# Peer review of "High Accurate Mathematical Tools to Estimate the Gravity Direction Using Two Non-Orthogonal Inclinometers"

_sensors, 2021, doi:10.3390/s21175727_

Round 1
Reviewer 1 Report
The article focuses on interesting and topical subject and is under the scope of the journal. It is well written, not too long and easy to read. It describes “Analytic Statistics of two Non-orthogonal Inclinometers” that makes it valuable to be published in the journal of Sensors. Only some improvements would be acknowledged clarifying some parts of the text, tables and figures, and updating some references. The following suggestions are provided to the author(s).
- The title is too short that is not appropriate to be itself. Please try to adapt it based on the journal guidelines and contents of manuscript.
- The abstract is quite confusing to me that is necessary to re-write. Please revise the abstract that the current one is too generic.
- Please improve the objectives and hypotheses of this study.
- Figures are not clear that should have higher resolution.
- In general, improving the quality of figures would help improve the readability.
- Conclusions should be included in a more concisely way and compared with similar studies. The discussion should be improved. There needs to be more comparative analysis with other studies.
- Please update the references that are a little outdated.
Reviewer 2 Report
I found this paper very interesting and also suitable for publication by Sensors after some relatively minor revisions.
My major comments include:
- In the second paragraph of Section 2, are the two measurement angles assumed to be independent (or conditionally independent) variables? Please clarify it.
- Moreover, in practice, are all the quantities ε, µx, µy and σ2 unknown? If so, why don’t comment about estimating them?
My minor comments include the following suggested changes and corrections:
Page 2, line 43. Please say the meaning of the acronym “PnP”.
Page 2, line 57. “... is derived by ...” instead of “... is derive by ...”.
Page 2, line 66. Please say the meaning of the acronym “WGS”.
Pages 8-9, Section 4. Which software was used to perform the Monte Carlo simulations? Please inform it.
Page 8, Figure 3. In the lower right panel, I think that it should be υy instead of υx in the x-axis.
Page 9, line 158. “... and a flattening factor (f),“ instead of “... and a flattening factor,“.
Page 10, line 179. “... Earth’s mass ...” instead of “... Earth’ mass ...”.
Page 11, line 195. In the argument of the double integral, shouldn’t it be Si and also Sj?
Page 13, line 230. In “... unbiased Gaussian errors.”, I think that it should be “zero-mean“ instead of “unbiased“ .
Round 2
Reviewer 1 Report
First of all, I appreciate to the authors for making efforts to carry out the changes by the referees. The authors did a proper revision for corresponding the comments and suggestions of the reviewers.